# Mechanical Properties of Nano SiO_2_ and Carbon Fiber Reinforced Concrete after Exposure to High Temperatures

**DOI:** 10.3390/ma12223773

**Published:** 2019-11-17

**Authors:** Linsong Wu, Zhenhui Lu, Chenglong Zhuang, Yu Chen, Ruihua Hu

**Affiliations:** Department of Civil Engineering, Yangtze University, Jingzhou 434000, China; wulinsong12@126.com (L.W.); luzhenhui_17216@163.com (Z.L.); chenglongzhuang123@163.com (C.Z.); hrh199806@163.com (R.H.)

**Keywords:** NSCFRC, mechanical properties, microstructure, residual properties, high temperatures

## Abstract

This study presents the key mechanical and residual properties after high-temperature of different Nano SiO_2_ carbon fiber-reinforced concrete (NSCFRC) mixtures. A total of seven NSCFRC mixtures incorporating 0%–0.35% of carbon fiber by volume of concrete and 0%–2% Nano SiO_2_ by weight of the binder were studied. The key mechanical properties such as compressive strength, tensile strength, and flexural strength of NSCFRC with 0.25% carbon fiber and 1% NS were 6.8%, 20.3%, and 11.7% higher than PC (0% CFs, 0% NS), respectively. Scanning Electron Microscopy (SEM) shows that Nano SiO_2_ reduced the internal porosity and increased the compactness of the concrete matrix. Furthermore, the experimental result demonstrates that NSCFRC can improve the mechanical properties of concrete after high-temperature and equations were obtained to describe the evolution of residual properties at elevated temperatures. Results suggested that the effect of carbon fibers on the residual properties of concrete after high-temperature is less than steel fiber and polypropylene fiber. It was also indicated that adding appropriate Nano SiO_2_ to concrete is an effective means to improve the residual performance after high-temperature.

## 1. Introduction

Concrete is one of the most widely used building materials, but its tensile strength is relatively low compared to compressive strength. This defect limits the application of concrete in many ways. As a common reinforcement method, fiber reinforcement is currently being considered more widely [1,2,3,4]. Carbon fiber-reinforced concrete (CFRC) has been used in many projects since the 1970s due to its good thermal conductivity, lightweight quality (low density), and high modulus of elasticity [4,5,6,7,8,9]. it can effectively improve the tensile strength, flexural strength, impact resistance and crack resistance. Carbon fiber concrete has also been widely studied as smart (self-monitoring) structural materials in recent years due to its electrical conductivity [10,11,12].

Similar to other flexible fibers, carbon fiber generally has no significant influence on the improvements of compressive strength of concrete and may decrease the compressive strength when the high-volume fractions. Previous studies have shown that the addition of 0.25–1 vol.% carbon fiber reduces the compressive strength of concrete by 15%–50% [13,14]. Furthermore, with the wide application of carbon fiber concrete, the fire resistance research of CFRC was very limited. At present, researchers pay more attention to steel fiber-reinforced concrete and polypropylene fiber-reinforced concrete. The melting and ignition points of polypropylene fiber are around 150 °C and 400–500 °C, respectively. Polypropylene fiber melts at higher temperatures to form a large number of small pores, which causes the internal pore connectivity to enhance, reduce the internal vapor pressure, and has much better resistance to thermal spalling. Steel fiber has high thermal conductivity and can effectively transfer high temperatures to the interior of concrete, reducing the temperature gradient of concrete. Carbon fiber has a higher melting point and ignition points (>1000 °C) than polypropylene fiber, and its thermal conductivity is lower than that of steel fiber. Therefore, the residual mechanical properties of carbon fiber concrete after high-temperature are worrying.

In order to improve the mechanical properties of CFRC after high-temperature, Nano SiO_2_ (NS) was added to CFRC to prepare NSCFRC. It is well documented that the mechanical and physical properties of concrete can be greatly improved by combining the pozzolanic nano materials such as SiO_2_ with concrete [15,16]. Due to the ultra-fine size of Nano SiO_2_, the formation of C-S-H gel nanocrystals can be accelerated and most of the pores of traditional cement-based concrete can be filled [17]. Formerly, the effect of the addition of silica nanoparticles on the mechanical properties of concrete was investigated. For example, Heidari [18] shown that the incorporation of nano-SiO_2_ particles can significantly improve the compressive, flexural, tensile strength and toughness of concreteIn addition, NS can improve the mechanical properties of concrete after high-temperature. Yan et al. [19] research showed that the addition of Nano SiO_2_ can evidently improve normal-temperature and high-temperature compressive strengths of the concrete.

This study presents the experimental results obtained on seven NSCFRC mixtures subjected to high temperatures. The main objective of this study was to investigate the effect of carbon fibers and Nano SiO_2_ on the mechanical properties at room temperature and high-temperature. 

The study is realized in two stages. Firstly, this paper supplies results on the mechanical properties of seven mixtures at room temperature. The effect of carbon fiber content and NS content on the mechanical properties and microstructure of concrete was revealed. Secondly, seven mixtures heated to 375 °C, 575 °C, 775 °C, and natural cooling. The evolution of residual mechanical properties with temperature were investigated in order to explain the influence of carbon fiber and NS on the mechanical properties of concrete after high temperature.

## 2. Experimental Program

### 2.1. Materials and Mixture Proportions

Normal Portland Cement (42.5R, Huaxin Cement Co., Ltd., Wuhan, China), Crushed Limestone Coarse Aggregate (CA), Natural Sand Fine Aggregate (FA), Nano Silica (NS) (A380, Evonik Industries AG, Essen, Germany), Water Reducer (WR) (F10, BASF SE, Ludwigshafen, Germany), Carbon Fibers (CFs) (T300, GW COMPOS, Weihai, China), Normal Tap Water (W), Dispersant (D) (Carboxyethylcellulose, HEC, Sinopharm Chemical Reagent Co., Ltd., Beijing, China) and Defoaming Agent (DA) (Tributyl Phosphate, AR, Sinopharm Chemical Reagent Co., Ltd., Beijing, China) were used in this study.

Table 1 shows the major properties of CFs and Nano SiO_2_. The diameter and length of the selected carbon fibers were 7 μm and 7 mm, respectively. Hence, the aspect ratio was 1000. The selected Nano SiO_2_ is Evonik A380 with specific surface area and particle size of 380 m^2^/g and 7 nm, respectively. The carbon fiber surface is hydrophobic and difficult to disperse uniformly in the cement paste, which directly affects the mechanical and other properties of the CFRC. Hydroxyethyl Cellulose (HEC, Sinopharm Chemical Reagent Co., Ltd., Beijing, China) was used as a dispersant in this study. Defoamer (Tributyl Phosphate, Sinopharm Chemical Reagent Co., Ltd., Beijing, China) was used to reduce pores generated during concrete mixing.

Table 2 presents the details of concrete mixture proportions. A total of seven NSCFRC mixtures incorporating 0%–0.35% carbon fibers by volume of concrete and 0%–2% Nano SiO_2_ by weight of the binder (C + NS) were studied. Nano SiO_2_ also improved the dispersion of carbon fibers in NSCFRC.

### 2.2. Preparation of Concretes

The carbon fiber and Nano SiO_2_ were pre-dispersed. The carbon fiber was placed in a beaker and a certain amount of water was added. The SiO_2_ and the dispersing agent being added, ultrasonically shaken and stirred for 10 min and the defoamer being added in this process reduces the bubbles generated during the mixing process. All dry components (cement and aggregate) were first pre-mixed together in the mixing machine (HJW-60, Kexin instrument Co., Ltd., Cangzhou, China) for 2 min. The dispersed carbon fiber and Nano SiO_2_, the remaining water and WR were added into the forced mixer, mixed the concrete mixture was mixed further for 2 min. Finally, concrete was poured into the molds and kept for 24 h, when they were demolded and conserved in curing room (25 °C and Relative Humidity > 95%) until 28 days of age.

### 2.3. Testing of Concretes

For each different concrete, 36 specimens were prepared and tested: 24 cubic specimens (150 mm × 150 mm × 150 mm) for the compression test and splitting tension test, and 12 prismatic specimens (100 mm × 100 mm × 400 mm) were made for the flexure test (Hongshan Testing Machine Co., Ltd., Tianshui, China). The purpose of this work was to evaluate the mechanical properties of NSCFRC after high temperatures. Therefore, the following tests were carried out under 4 different conditions (25, 375, 575 and 775 °C). For each different concrete 27 specimens included 18 cubic specimens and nine prismatic specimens were heated by box type resistance furnace (Jianli Co., Ltd., Huanggang, China) at 375, 575, and 775 °C. Once reached the target temperature, it was maintained for 120 min to obtain a homogeneous thermal state within the specimens. After that, turn off the oven and store the sample inside for natural cooling All samples were tested after cooling up to ambient temperature.

The compressive strength, splitting tensile strength, and flexure strength of NSCFRC and control mixtures were determined according to GB/T50081-2016.

The microstructure of the fractured concrete surface was observed by Scanning Electron Microscopy (SEM, TESCAN, Brno, Czech) 

## 3. Results and Discussion

### 3.1. Mechanical Properties at Room Temperature

#### 3.1.1. Compressive Strength

The compressive strength results and associated standard deviation (SD) for different NSCFRC as show in Table 3. All data performed one sample *t*-test with 95% confidence, the population mean is not significantly. The experimental results were statistically acceptable. Figure 1a shows that specimen C2 (0.25% carbon fiber, 1% Nano SiO_2_) provided the highest compressive strength of 47.1 MPa, 6.8% higher than PC (0% carbon fiber and 0% Nano SiO_2_). However, the compressive strength of other specimens generally declined. C3 (0.35% carbon fiber and 1% Nano SiO_2_) provided the lowest compressive strength of 39.9 MPa. Comparing with PC (0% carbon fiber, 0% Nano SiO_2_) and S0 (0.15% carbon fiber, 0% Nano SiO_2_), the compressive strength of carbon fibers decreased by 15.5%. Carbon fibers had a negative effect on the compressive strength of concrete. The compressive strength of NSCFRC was between 81.0 and 106.5% of PC in the presence of carbon fibers. A previous study reported that carbon fibers significantly decreased the compressive strength of CFRC, because it increased internal porosity and reduced the compactness of the concrete matrix [20]. Other studies have shown that carbon fibers are not as strong as aggregates to resist compression [21].

Appropriate amount of carbon fiber and NS can improve the compressive strength of concrete. C2 (0.25% carbon fiber, 1% Nano SiO_2_) had the highest compressive strength, which proved that Nano SiO_2_ can effectively improve the compressive strength of concrete. Furthermore, the compressive strength of the specimens increased with the increase in Nano SiO_2_ content. The compressive strength for S0 (0.15% carbon fiber and 0% Nano SiO_2_) was 37.3 MPa and for S2 (0.15% carbon fiber and 1% Nano SiO_2_) was 41.2 MPa. The compressive strength increased by 10% for 1% Nano SiO_2_, since it reduced the internal porosity and increased the compactness of the concrete matrix. The study by Wang et al. [22] reported that the addition of 3% Nano SiO_2_ significantly increased the compressive strength of Lightweight aggregate concrete (LWAC). Cheng et al. [23] reported that the incorporation of 1% Nano SiO_2_ greatly improved the strength, and abrasion resistance of concrete. Zhang et al. [24] reported that when the content of nano-SiO_2_ is limited to a certain range, the incorporation of concrete. Therefore, in this experiment, the optimal content of concrete to compressive strength was kept at 1 wt.% NS and 0.25 vol.% CFs.

#### 3.1.2. Splitting Tensile Strength

The splitting tensile strength results and associated standard deviation (SD) for different NSCFRC as show in Table 3. All data performed one sample t-test with 95% confidence, the mean is not significantly. The experimental results were statistically acceptable. Figure 1b show that the splitting tensile strength of NSCFRC increased first and then decreased as the carbon fiber content increased. PC had the tensile strength of 2.76 MPa while C3 (0.25% carbon fiber and 1% Nano SiO_2_) had the tensile strength of 3.32 MPa, the splitting tensile strength increased by 20% for 0.25% carbon fiber, 1% Nano SiO_2_. Table 3 shows the effect of carbon fibers and Nano SiO_2_ on the splitting tensile strength of concrete. The tensile strength of NSCFRC increased first. Many articles reported a similar trend in the tensile strength results of CFRC, a higher load is obtained because the increase in carbon fiber content reduces the crack growth [13,20]. When the carbon fiber content reached 0.35%, the carbon fiber content is too high to be uniformly dispersed causing defects in concrete to increase and the tensile strength to decrease.

The tensile strength of NSCFRC decreased first and then increased as the Nano SiO_2_ content increased. S0 (0.15% carbon fiber, 0% Nano SiO_2_) had a tensile strength of 2.98 MPa while S3 (0.15% carbon fiber, 2% Nano SiO_2_) had a tensile strength of 3.24 MPa, the splitting tensile strength increased by 8.7% for 2% Nano SiO_2_. 

Nazari et al. [25] reported that when SiO_2_ is less than 3%, the formation of C-S-H gel can be accelerated due to the increase of crystalline Ca (OH)_2_ content in the initial stage of hydration reaction, thereby improving the tensile strength of concrete specimens. When the SiO_2_ content is more than 3%, the reduction of the tensile strength is caused by the decrease of the crystalline Ca(OH)_2_ content required for the formation of the C-S-H gel. Other researchers have reported similar content, but the optimal dosage of NS was different [26,27]. NS can react with C-H in cement to form C-S-H, effectively filling the concrete pores and increasing the matrix compactness. It effectively improved the splitting tensile strength of concrete. Therefore, in this work, the optimal content of concrete for splitting tensile strength was 2% NS.

#### 3.1.3. Flexural Strength

The flexural strength results and associated standard deviation (SD) for different NSCFRC as show in Table 3. All data performed one sample t-test with 95% confidence, the mean is not significantly. The experimental results were statistically acceptable. Figure 1c shows that the flexural strength of C1, C2 and C3 was 3.85MPa, 4.10Mpa, and 4.30MPa, respectively. The flexural strength of C3 about 19% higher than PC. The flexural strength for the specimens increased as the carbon fiber content increased because the carbon fibers bridged the cracks and significantly reduced the crack opening, thus increasing the flexural strength of concrete. The increased number of fibers that cross the crack surface also increases the flexural strength of concrete [20,28].

Figure 1c also shows that the flexural strength increased first and then decreased as the SiO_2_ content increased. The flexural strength ranged from 3.57 to 3.85 MPa. S2 (0.15% CFs and 1% NS) had the highest flexural strength of 3.85 MPa while S3 (0.15% CFs and 2% NS) had the lowest flexural strength of 3.57 MPa. The flexural strength of S2 was 6.4% higher than that of S0 (0.15% CFs, 0% NS). The results indicated that Nano SiO_2_ were able to improve the flexural strength of concrete due to the Pozzolanic and filling effects. A previous study reported that the increased SiO_2_ nanoparticles content (more than 4 wt. %) causes reduced flexural strength due to unsuitable dispersion of nanoparticles in the concrete matrix [25]. This was also the main reason for the lowest flexural strength of specimen S3.

#### 3.1.4. Microstructure of NSCFRC

Figure 2 shows the microstructure of the fracture surface for different NSCFRCs observed via SEM. The SEMs exhibited that the carbon fibers were uniform dispersion in concretes without agglomeration, proving that the carbon fiber content can be evenly dispersed in concrete when it is 0.15 vol.% of concrete. 

Figure 2d shows the microstructure of specimen C2 with 0.25% carbon fiber and 1% Nano SiO_2_, which provided the highest mechanical properties due to compact matrix and evenly distributed carbon fiber. 

Figure 3 shows the two fiber failure modes (fiber pullout and fiber breakage), observed by SEM. Safiuddin et al. [20] found that the pitch-based carbon fibers were pulled out or broken after the peak loading. In addition, Chen et al. [28] also observed fiber pull-out and fiber breakage in CFRC. Figure 3 also shows the clustering and uneven distribution of carbon fiber, which might be the reason for the decrease in mechanical properties of specimen C3.

### 3.2. Residual Compressive Strength

Table 4 shows the compressive strength of NSCFRC after exposure to high temperatures. Normalized residual compressive strength is the ratio of the original compressive strength at room temperature after exposure and cooling to high temperature. All data performed one sample t-test with 95% confidence, the mean is not significantly. The experimental results were statistically acceptable; however, the standard deviation of the results indicates that the data are more discrete as the temperature increases. Especially for S1, the standard deviation is above 30% at 775 °C.

Figure 4a show that the compressive strength of concrete decreases with increasing temperature. The compressive strength decreased slightly at 375 °C and the normalized residual strength of all samples were greater than 88%. The normalized residual strength ranges from 60 to 69% at 575 °C, and the normalized residual strength is less than 45% at 775 °C. It can be seen that the presence of NS can effectively increase the residual compressive strength of concrete after high-temperature. After 775 °C, the compressive strength of S3 (0.15% CFs and 2% NS) is 17.3 MPa, which is lower than that of S2 (0.15% CFs and 1% NS) at 18.0 MPa. However, S3 provided the highest normalized residual compressive strength of 45.3%, 7.5% higher than S0 (0.15% CFs and 0% NS), and 10.2% higher than PC (0% CFs and 0% NS).

Figure 4b shows the relationship between the normalized residual compressive strength and temperature. Figure 4b compared the results with other curves from previous studies. Included the Eurocode 2 curve with siliceous aggregate and without fibers [29]. NSC with hybrid fiber by Varona [30], steel fiber reinforced concrete (SFRC) by Li [31] and polypropylene fibers reinforced concrete (PPFRC) by Aslani et al. [32] and Ding et al. [33]. It can be seen that NSCFRC gave better residual strength than the EC-2 and polypropylene fibers reinforced concrete from Aslani et al. and Ding et al. curve, included PC (0% CFs and 0% NS). The reason for it may not only the incorporation of CFs and NS, but also used calcium aggregates, which perform better than siliceous aggregates under high temperature. Quartz in siliceous aggregate will phase change at a high temperature of about 575℃, Causes the aggregate volume to expand, resulting generates cracks and weakening of the concrete matrix. The residual compressive strength of NSCFRCs is also higher than PPFRC by Aslani et al. and Ding et al. because the residual compressive strength reduces when the temperature is above 375 °C due to the fiber has been melted up at such high temperature and the pores left are disadvantageous for the performance of concrete [32,33]. Moreover, experimental results were similar to the Varona curve for hybrid fiber reinforce concrete. However, the normalized residual compressive strength of NSCFRC is lower than that of steel fiber reinforced concrete by Li et al. This would support the incorporation of carbon fiber and NS to increase the normalized residual compressive strength of concrete. 

The experimental results were worked to design equations by the means of regression analyses. Equation (1) corresponds to NSCFRC with 0.25% CFS and 1% NS, with a coefficient of determination R^2^ = 0.98175:(1)$$\frac{f_{c,T}}{f_{c,25}}=1.0048-1.166\times 10^{-5}T-1.057\times 10^{-7}T^{2}\;(25\;{}^{\circ}C<T<775\;{}^{\circ}C),$$
where $f_{c,T}$ is the residual compressive strength of concrete after exposure to high temperature *T*; *f_c_*_,25_ is the compressive strength of concrete before exposure to high temperature; *T* is the elevated temperature experienced by the concrete (unit: °C).

### 3.3. Residual Tensile Strength

Table 5 shows the tensile strength of NSCFRC after exposure to high temperatures. All data performed one sample t-test with 95% confidence, the mean is not significantly. The experimental results were statistically acceptable the standard deviation of the results indicates that the data is more discrete as the temperature increases.

Figure 5 represents the evolution of the tensile strength of NSCFRC after high temperatures. It can be seen that the tensile strength of concrete decreases with increasing temperature. The tensile strength has a certain decrease after 375 °C, and the normalized residual tensile strength is between 82% and 89%. When the temperature exceeds 375 °C, the normalized residual tensile strength decreases dramatically, and the normalized residual tensile strength is between 26% and 37% after 775 °C. After exposure to high temperature, the normalized residual tensile strength of NSCFRC improved compared to PC due to the presence of carbon fiber and NS. After 775 °C, the normalized residual tensile strength of S2 (0.15% CFs and 0% NS) was 3.5% higher than S0 (0.15% CFs and 1% NS), and the residual tensile strength of C2 (0.25% CFs and 1% NS) was 10.8% higher than PC.

In Figure 5b, the experimental results for tensile strength were compared with other curves from previous studies. Included the Eurocode 2, the NSC with hybrid fiber by Varona et al. [30], steel fiber reinforced concrete by Li et al. [31], polypropylene fibers reinforced concrete by Aslani et al. [32], and HSC with hybrid fiber by Gao [34]. The experimental results showed that the residual tensile strength of NSCFRC was higher than that of EC-2, there is a significant difference between EC-2 and both the results and other curves. However, Figure 5b illustrates that the NSCFRC and other curves still have good residual tensile strength after the temperature exceeds 575 °C. The reason for such improvements could be attributed to the calcareous aggregates and fiber. Experimental results were similar to the prediction in Gao curve for HSC with hybrid fiber below 575 °C. The reason for it may be that NSC and NSCFRC had less internal porosity. However, at 775 °C, the residual tensile strength of NSCFRC is lower than Gao curve because carbon fiber has a deterioration of mechanical properties due to oxidation, and it cannot provide sufficient tensile strength above 575 °C. Therefore, steel fiber reinforced concrete has the highest residual tensile strength at 775 °C.

The experimental results were worked to design equations by the means of regression analyses. Equation (2) corresponds to NSCFRC with 0.25% CFs and 1% NS, with a coefficient of determination R^2^ = 0.9999
(2)$$\frac{f_{ct,T}}{f_{ct,25}}=1/\left(1+8.21\times 10^{-10}{\left(T-25\right)}^{3.24}\right)\;(25\;{}^{\circ}C<T<775\;{}^{\circ}C),$$ where $f_{ct,T}$ is the residual tensile strength of concrete after exposure to high temperature *T*; *f_ct_*_,25_ is the tensile strength of concrete before exposure to high temperature; *T* is the elevated temperature experienced by the concrete (unit: °C).

### 3.4. Residual Flexural Strength

Table 6 shows the flexural strength of NSCFRC after exposure to high temperatures. All data performed one sample t-test with 95% confidence, the mean is not significantly. The experimental results were statistically acceptable.

Figure 6a can be seen that the flexural strength of concrete decreases with increasing temperature. The flexural strength of concrete decreases by about 30% at 375 °C, 60% at 575 °C, and 80% at 775 °C. After exposure to high temperatures, the flexural strength of NSCFRC improved compared to PC. After 575 °C, specimen C3 (0.25% carbon fiber, 1% Nano SiO_2_) provided the highest flexural strength of 1.86 MPa, and was 40% higher than that of PC. The normalized residual flexural strength of C3 was 6% higher than that of PC. After 775 °C, C3 (0.35% CFs, 1% NS) had the highest flexural strength while PC had the lowest flexural strength. The normalized residual flexural strength of C3 was 10% higher than PC.

Figure 6a illustrates that the carbon fibers had no significant effect on the normalized residual flexural strength of concrete. The normalized residual flexural strength of concrete with different carbon fiber content after exposure to high temperatures had little difference. However, the NS content had a certain influence on the normalized residual flexural strength of concrete. Compared between S0 and S3, the normalized residual flexural strength increased by 6.2%.

Figure 6b shows the experimental results compared with other models, including the NSC with hybrid fiber by Varona et al. [30], steel fiber reinforced concrete by Li et al. [31], polypropylene fibers reinforced concrete, and steel fibers reinforced concrete by Aslani et al. [32,35]. All curves were linear and similar. Nonetheless, there seems to be no need for providing additional equations for the flexural strength of NSCFRC.

### 3.5. Mechanism of Mechanical Properties of NSCFRC after High Temperature

From the previous section, it can be seen that the NSCFRC has better residual mechanical properties compared to the concrete without fiber and polypropylene fiber reinforced concrete but lower than steel fiber reinforced concrete. Polypropylene fiber had a low melting point and melted at high temperatures, forming a large number of small pores, which caused the internal pore structure of the concrete to change. The pore connectivity was strengthened, providing a channel for the decomposition and evaporation of moisture inside the concrete, which greatly reduced the internal pressure; thereby, Polypropylene fiber reinforced concrete has much better resistance to thermal spalling compared to the concrete without fiber. Steel fiber had high thermal conductivity and can effectively transfer high temperature to the interior of concrete, reducing the temperature gradient of concrete [36]. Carbon fiber has a high melting point and a burning point (>1000 °C); the testing temperatures are not high enough to melted carbon fiber, Therefore, its ductility can effectively contribute to the concrete resisting tensile damage, but carbon fiber has a deterioration of mechanical properties due to oxidation above 575 °C.

NS can effectively improve the mechanical properties of concrete after high-temperature. Comparing S0 to S3, after 775 °C, the normalized residual compressive strength, residual tensile strength, and residual flexural strength increased by 7.5, 3.5, and 6.3%, respectively. NS can react with C-H in cement to form C-S-H, and effectively fill concrete pores and increase the matrix compactness. It effectively alleviates the deterioration of concrete at high temperatures [37].

## 4. Conclusions

This study reports the experimental study on the mechanical properties of NSCFRC after room-temperature and high-temperature. Based on the test results and associated discussion, the following conclusions were drawn:
NSCFRC provided relatively good mechanical properties than PC. The compressive strength, tensile strength and flexural strength of NSCFRC with 0.25% carbon fiber and 1% NS were 6.8%, 20.3% and 11.7% higher than that of PC, respectively.The existence of NS can comprehensively improve the mechanical properties of concrete. The compressive strength, tensile strength and flexural strength of S2 (0.15% CFs and 1% NS) were 10%, 6.7%, and 6.4% higher than that of S0 (0.15% CFs without NS), respectively, since it reduces the internal porosity and increases the compactness of the concrete matrix. NS can react with C-H in cement to form C-S-H, which reduces the internal porosity and increases the matrix compactness.Similar to previous reports, carbon fiber substantially decreased the compressive strength of CFRC because it increased the internal porosity and reduced the compactness of the concrete matrix. However, it can increase the tensile strength and flexural strength of concrete because it reduces the crack growth across the crack surface, thus resulting in failure at a higher load.NSCFRC can improve the mechanical properties of concrete after high-temperature. The effect of high temperature on the residual mechanical properties of NSCFRC was less than in PC, but greater than that of steel fiber-reinforced concrete. After 775 °C, the normalized residual compressive strength, residual tensile strength, and residual flexural strength of NSCFRC with 0.25% carbon fiber and 1% NS were 5.2%, 10.9%, and 8.9% higher than those of PC, respectively.NS can effectively improve the mechanical properties of concrete after high-temperature. NS can reduce the internal porosity and increase the matrix compactness after high-temperature. The synergistic effect of NS and carbon fiber was the main factor for the improvement of mechanical properties of NSCFRC after high-temperature.

## Figures and Tables

**Figure 1 materials-12-03773-f001:**
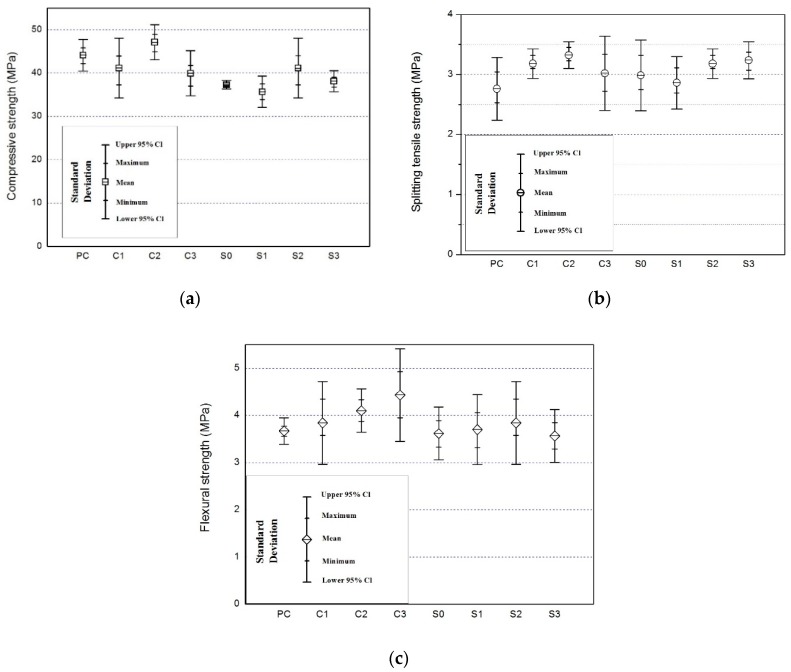
(**a**) The compressive strength of NSCFRC; (**b**) tensile splitting strength of NSCFRC; (**c**) The flexural strength of NSCFRC.

**Figure 2 materials-12-03773-f002:**
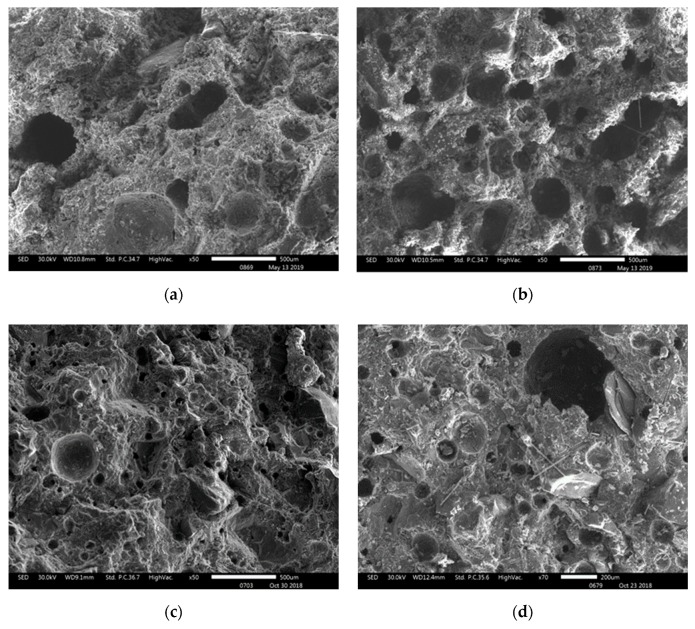
SEM of fracture surface of NSCFRC, (**a**) PC (0% CFs, 0% NS); (**b**) S0 (0.15% CFs, 0% NS); (**c**) S2 (0.15% CFs, 1% NS); (**d**) C2 (0.25% CFs, 1% NS).

**Figure 3 materials-12-03773-f003:**
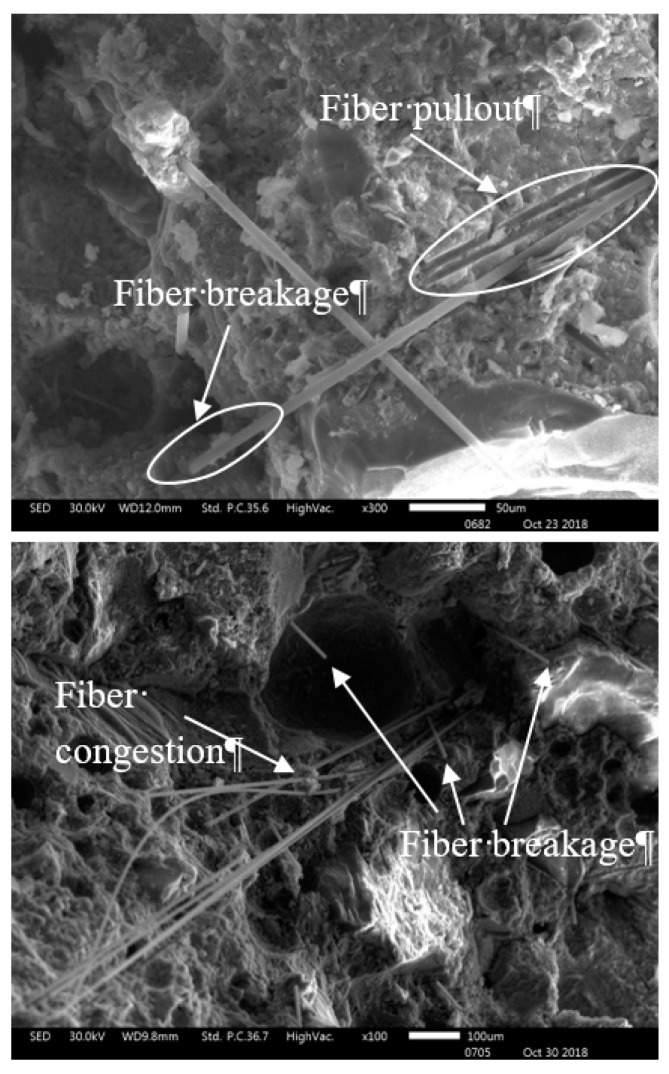
SEM of fracture surface of C3.

**Figure 4 materials-12-03773-f004:**
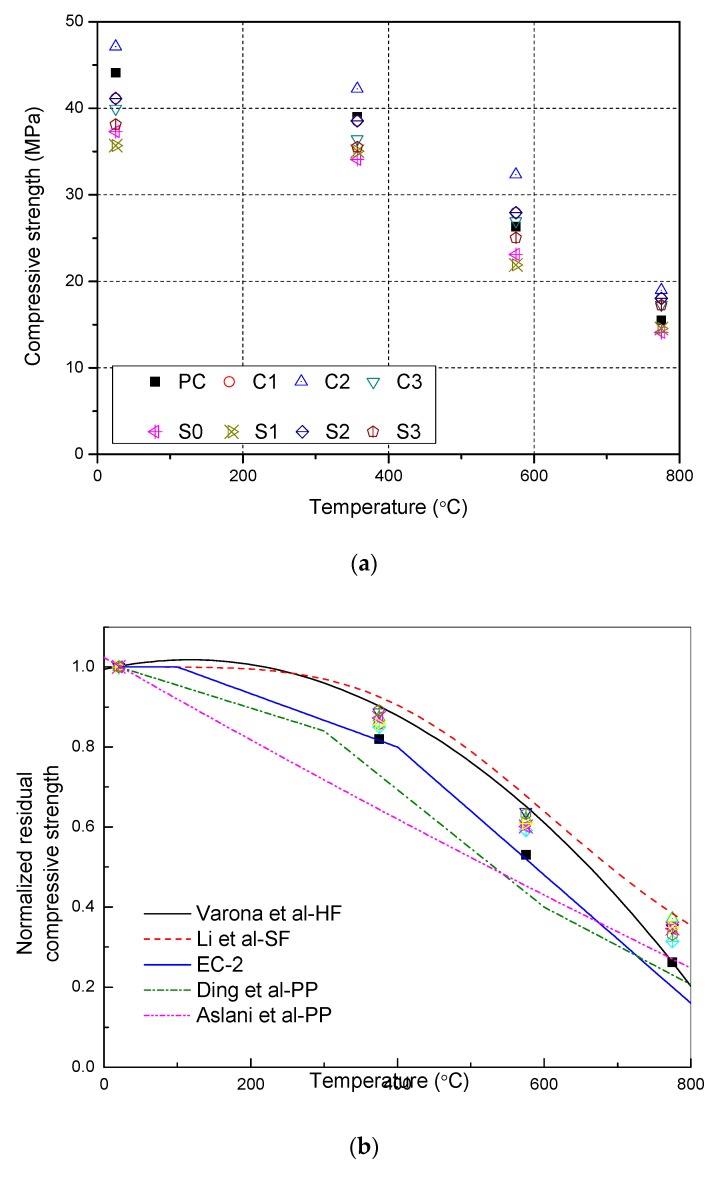
Evolution of the NSCFRC compressive strength after high temperatures; (**a**) The residual compressive strength after exposure to high temperature; (**b**) The normalized residual compressive strength after exposure to high temperature.

**Figure 5 materials-12-03773-f005:**
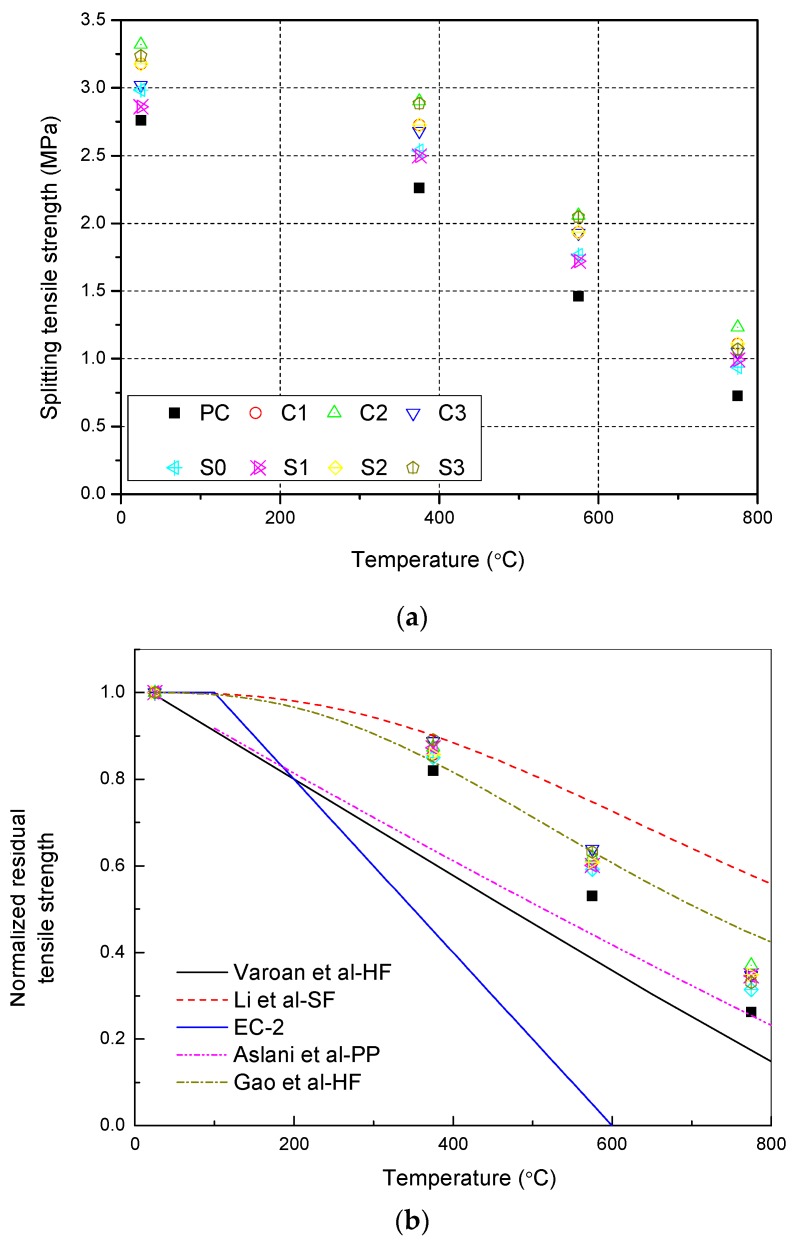
Evolution of the NSCFRC tensile strength at high temperatures: (**a**) The residual tensile strength after exposure to high temperature; (**b**) The normalized residual tensile strength after exposure to high temperature.

**Figure 6 materials-12-03773-f006:**
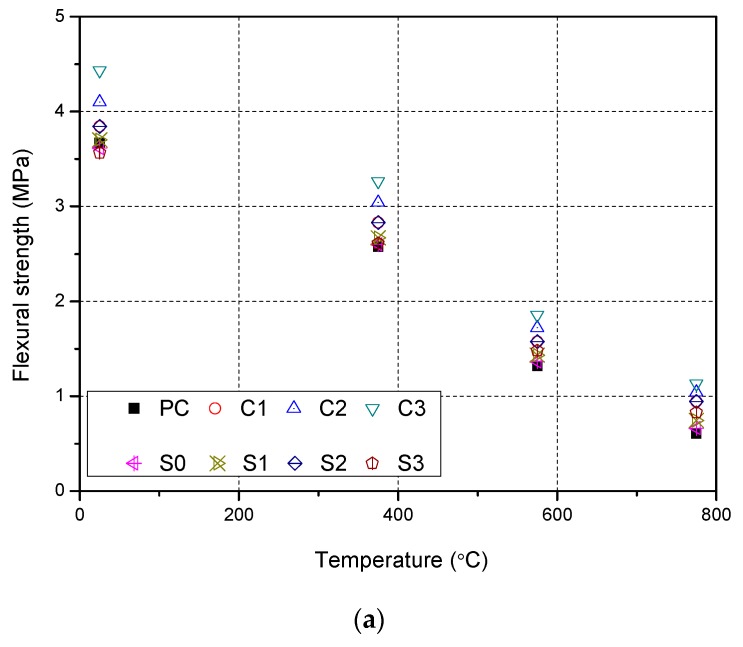
Evolution of the NSCFRC flexural strength at high temperatures: (**a**) The residual flexural strength after exposure to high temperature; (**b**) The normalized residual flexural strength after exposure to high temperature.

**Table 1 materials-12-03773-t001:** Major properties of carbon fibers and nano-SiO_2_.

Material	Properties
**Carbon fibers**	Tensile strength: 3530 MPa Tensile modulus:230 GPa
	Length: 7 mm Diameter: 7 μm Volume density: 1.76 g/cm^3^ Carbon content: 93%
**Nano-SiO_2_**	Specific surface area: 380 ± 30 m^2^/g Mean Particle size: 7 nm
	Apparent density: ≈ 30 wt.%
	Bulk density: ≈ 50 wt.%
	SiO_2_ content: > 99.8%

**Table 2 materials-12-03773-t002:** Details of mixture proportions per unit volume of concrete.

Concrete Mixture	Water/Binder Ratio	Water (kg/m^3^)	Cement (kg/m^3^)	FA (kg/m^3^)	CA (kg/m^3^)	Nano SiO_2_ (kg/m^3^)	CFs (kg/m^3^)	D (wt.% of Binder)	WR (wt.% of Binder)	DA (wt.% of Binder)
**PC**	0.45	193.5	430.0	710	1075	0	0	0.6	1.0	0.6
**C1**	0.45	193.5	425.7	710	1075	4.3	2.58	0.6	1.25	0.6
**C2**	0.45	193.5	425.7	710	1075	4.3	4.3	0.6	1.5	0.6
**C3**	0.45	193.5	425.7	710	1075	4.3	6.02	0.6	1.75	0.6
**S0**	0.45	193.5	430.0	710	1075	0	2.58	0.6	1.0	0.6
**S1**	0.45	193.5	427.9	710	1075	2.1	2.58	0.6	1.25	0.6
**S2**	0.45	193.5	425.7	710	1075	4.3	2.58	0.6	1.25	0.6
**S3**	0.45	193.5	421.4	710	1075	8.6	2.58	0.6	1.25	0.6

PC (0% CFs, 0% NS); C1(0.15% CFs, 1% NS); C2 (0.25% CFs, 1% NS); C3 (0.35% CFs and 1% NS); S0 (0.15% CFs, 0% NS); S1(0.15% CFs and 0.5% NS); S2 (0.15% CFs and 1% NS); S3 (0.15% CFs and 2% NS).

**Table 3 materials-12-03773-t003:** Compressive strength, Splitting tensile strength and flexural strength of different NSCFRC.

Concrete Mixture	CFs (vol.% of Concrete)	Nano SiO_2_ (wt.% of Binder)	Compressive Strength (MPa)	Tensile Strength (MPa)	Flexural Strength (MPa)
**PC**	0	0	44.1 (4.1%)	2.76 (9.4%)	3.67 (3.8%)
**C1**	0.15	1	41.2 (8.4%)	3.18 (3.9%)	3.85 (11.4%)
**C2**	0.25	1	47.1 (4.3%)	3.32 (3.4%)	4.10 (5.6%)
**C3**	0.35	1	39.9 (6.5%)	3.02 (10.2%)	4.43 (11.1%)
**S0**	0.15	0	37.3 (1.3%)	2.98 (9.9%)	3.62 (7.8%)
**S1**	0.15	0.5	35.7 (7.2%)	2.86 (7.6%)	3.70 (10.0%)
**S2**	0.15	1	41.2 (8.4%)	3.18 (3.9%)	3.85 (11.4%)
**S3**	0.15	2	38.1 (3.1%)	3.24 (4.7%)	3.57 (7.9%)

**Table 4 materials-12-03773-t004:** Compressive strength of different NSCFRC after high-temperature.

Concrete Mixture	CFs (vol.% of Concrete)	Nano SiO_2_ (wt.% of Binder)	25 °C (MPa)	375 °C (MPa)	575 °C (MPa)	775 °C (MPa)
**PC**	0	0	44.1 (4.1%)	39.0 (5.4%)	26.3 (10.4%)	15.5 (26.6%)
**C1**	0.15	1	41.2 (8.4%)	38.5 (8.7%)	27.9 (11.8%)	18.0 (14.1%)
**C2**	0.25	1	47.1 (4.3%)	42.3 (6.0%)	32.3 (16.0%)	19.0 (14.3%)
**C3**	0.35	1	39.9 (6.5%)	36.4 (8.9%)	27.0 (8.1%)	17.3 (24.0%)
**S0**	0.15	0	37.3 (1.3%)	34.1 (4.9%)	23.1 (7.4%)	14.1 (7.0%)
**S1**	0.15	0.5	35.7 (7.2%)	35.0 (2.5%)	21.9 (6.7%)	14.6 (37.4%)
**S2**	0.15	1	41.2 (8.4%)	38.5 (8.7%)	27.9 (11.8%)	18.0 (14.1%)
**S3**	0.15	2	38.1 (3.1%)	35.5 (13.3%)	25.0 (18.2%)	17.2 (4.0%)

**Table 5 materials-12-03773-t005:** Splitting tensile strength of different NSCFRC after high-temperature.

Concrete Mixture	CFs (vol.% of Concrete)	Nano SiO_2_ (wt.% of Binder)	25 °C (MPa)	375 °C (MPa)	575 °C (MPa)	775 °C (MPa)
**PC**	0	0	2.76 (9.4%)	2.26 (8.9%)	1.46 (16.1%)	0.72 (28.2%)
**C1**	0.15	1	3.18 (3.9%)	2.73 (4.0%)	1.94 (5.0%)	1.11 (18.0%)
**C2**	0.25	1	3.32 (3.4%)	2.90 (4.6%)	2.06 (2.3%)	1.23 (7.9%)
**C3**	0.35	1	3.02 (10.2%)	2.68 (7.4%)	1.93 (12.9%)	1.05 (17.7%)
**S0**	0.15	0	2.98 (9.9%)	2.54 (5.6%)	1.77 (10.7%)	0.94 (15.9%)
**S1**	0.15	0.5	2.86 (7.6%)	2.50 (5.3%)	1.72 (27.0%)	0.99 (13.9%)
**S2**	0.15	1	3.18 (3.9%)	2.73 (4.0%)	1.94 (5.0%)	1.11 (18.0%)
**S3**	0.15	2	3.24 (4.7%)	2.88 (6.1%)	2.05 (5.2%)	1.07 (25.0%)

**Table 6 materials-12-03773-t006:** Flexural strength of different NSCFRC after high-temperature.

Concrete Mixture	CFs (vol.% of Concrete)	Nano SiO_2_ (wt.% of Binder)	25 °C (MPa)	375 °C (MPa)	575 °C (MPa)	775 °C (MPa)
**PC**	0	0	3.67 (3.8%)	2.58 (6.5%)	1.32 (18.3%)	0.61 (7.4%)
**C1**	0.15	1	3.85 (11.4%)	2.83 (8.6%)	1.57 (5.2%)	0.94 (11.8%)
**C2**	0.25	1	4.10 (5.6%)	3.04 (4.1%)	1.72 (17.4%)	1.04 (8.0%)
**C3**	0.35	1	4.43 (11.1%)	3.26 (5.5%)	1.86 (8.3%)	1.13 (12.5%)
**S0**	0.15	0	3.62 (7.8%)	2.60 (16.8%)	1.36 (14.5%)	0.66 (14.5%)
**S1**	0.15	0.5	3.70 (10.0%)	2.67 (15.4%)	1.43 (23.4%)	0.74 (21.2%)
**S2**	0.15	1	3.85 (11.4%)	2.83 (8.6%)	1.57 (5.2%)	0.94 (11.8%)
**S3**	0.15	2	3.57 (7.9%)	2.61 (17.7%)	1.48 (9.8%)	0.83 (21.8%)

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
