# Peer review of "Mechanical Properties of Nano SiO2 and Carbon Fiber Reinforced Concrete after Exposure to High Temperatures"

_materials, 2019, doi:10.3390/ma12223773_

Round 1

Reviewer 1 Report

Overall an interesting study on the use of carbon fibers and nanosilica for high temperature concrete behavior. The experimental methodology is sound and the authors clearly present the data. A number of comments are highlighted below detailing a number of recommendations pertaining to adding more detail or discussion to the article. The article is readable, but minor editing for proper English should be performed. 

-Introduction. "Similar to other flexible fibers, carbon fiber has a detrimental effect on the compressive properties of concrete." Does this statement refer to high volume fractions of fibers? In my work, I don't see much of a strength loss at low fiber volumes. Perhaps this statement can be reworded to be less definitive. 

-Introduction. "Previous studies have shown that the addition of carbon fiber reduces the compressive strength of concrete by 15%–50%..." Like the previous comment, at what volume fraction is this reduction in strength? Please specify. 

-Introduction. "Compared with carbon fiber, polypropylene  fiber melt at higher temperatures to form a large number of small pores" At what temperatures does this happen? At what temperature is carbon fiber susceptible for fire damage?

-Section 2.1. Do the authors mean "natural sand" instead of "nature sand"?

-Section 2.1. What is the purpose of the defoaming agent?

-Table 1. For the nanosilica particle size, does the 7nm refer to a mean size, median size, etc? Are all the particle 7nm or is there a size distribution?

-Section 2.3. How many replicate specimens were prepared and tested?

-Section 3.1.1, 3.1.2, and 3.1.3: please comment on the relative statistical variability in the data. Based on the results in Table 1, considering the standard deviation, are the results statistically different (for instance, according to a t-test with 95% confidence)? Including a discussion of the statistical variability would significantly bolster the arguments and the conclusions. 

-Section 3.1.4. The discussion on the porosity derived by SEM imaging of fracture surface should be revised. The authors are drawing too many conclusions based on one image. Yes, the SEM image qualitatively shows more pores, but are the authors certain that what they are interpreting as pores is not a result of the testing? Consider revising this discussion, or provide more definitive analysis of the porosity, such as by MIP, nitrogen adsorption, or image analysis of backscattered electron images of polished sections.

-Section 3.2, 3.3, and 3.4. Same comment as above regarding statistical analysis. By the variability reported in Table 4, 5, and 6, are the results statistically significant? Did nanosilica or fibers have a statistically relevant effect? Consider at least a t-test with 95% confidence. 

-Equation 2. There is a missing parenthesis in the equation.

-Section 3.5. "The decomposition of Ca(OH)2 has no critical influence on the reduction of concrete strength." If Ca(OH)2 converts to CaO + H2O, doesn't CaO have a lower unit volume than Ca(OH)2? If so, decomposition of Ca(OH)2 should increase porosity and therefore reduce strength.

-Section 3.5. "C–S–H starts to decompose at around 560°C and it decomposes into b-C2S at around 600–700°C." The authors may wish to check this reference. I've never heard of C-S-H converting to belite. This doesn't make sense to me. 

-Section 3.5. The authors should also comment on the influence of temperature on the aggregates. If limestone or dolomite is present, then they would undergo decarbonation at higher temperatures. If quartz is present, then it would undergo a phase transition at 573C and result in cracking. Consider reporting on the minerals present in the aggregates and how this is affected by temperature. 

-Section 3.5. "Compared with steel fiber and polypropylene fiber, carbon fiber had clear disadvantages in improving the mechanical properties of concrete after high-temperature." THis sentence is confusing. Did carbon fibers help or not?

-Section 3.5. Discuss what temperatures polypropylene fibers will melt. Are there any temperature-related effects for carbon fibers?

-Conclusions. For Conclusion #5, I am confused with regard to carbon fiber performance at high temperatures. Were carbon fibers useful or not at high temperatures? Didn't melt?

Author Response

Dear Reviewers:

    Thank you for your letter and for the reviewers’ comments concerning our manuscript entitled “Mechanical Properties of Nano SiO2 and Carbon Fiber Reinforced Concrete after Exposure to High Temperatures” (ID: materials-622485). Those comments are all valuable and very helpful for revising and improving our paper, as well as the important guiding significance to our researches. We have studied comments carefully and have made correction which we hope meet with approval. Revised portion a using the "Track Changes" function in paper. The main corrections in the paper and the responds to the reviewer’s comments are as flowing:

Introduction. "Similar to other flexible fibers, carbon fiber has a detrimental effect on the compressive properties of concrete." Does this statement refer to high volume fractions of fibers? In my work, I don't see much of a strength loss at low fiber volumes. Perhaps this statement can be reworded to be less definitive. 

Response 1 : We have made correction according to the Reviewer’s comments, this section is modified that “carbon fiber generally has no significant influence on the improvements of compressive strength of concrete and may decreased the compressive strength when the high-volume fractions”

Introduction. "Previous studies have shown that the addition of carbon fiber reduces the compressive strength of concrete by 15%–50%..." Like the previous comment, at what volume fraction is this reduction in strength? Please specify. 

Response 2: Line 40-41, the statements were corrected as “Previous studies have shown that the addition of 0.25-1 vol. % carbon fiber reduces the compressive strength of concrete by 15%–50%”

Introduction. "Compared with carbon fiber, polypropylene  fiber melt at higher temperatures to form a large number of small pores" At what temperatures does this happen? At what temperature is carbon fiber susceptible for fire damage?

Response 3: The melting and ignition points of polypropylene fiber are around 150 ℃ and 400–500 ℃, respectively.

Carbon fiber has higher melting point and ignition points (>1000 ℃), but some articles reported that carbon fiber has a deterioration of mechanical properties due to oxidation, and it cannot provide sufficient tensile strength above 600℃.

We add this discussion in line 44-45, 50-51, 326-330, 404-408.

Section 2.1. Do the authors mean "natural sand" instead of "nature sand"?

Response 4: We corrected this word

Section 2.1. What is the purpose of the defoaming agent?

Response 5: Line 98, The carbon fiber and Nano SiO2 will generate bubbles during mixing, and the defoamer was added in this process reduces the bubbles generated during the mixing process

Table 1. For the nanosilica particle size, does the 7nm refer to a mean size, median size, etc? Are all the particle 7nm or is there a size distribution?

Response 6: We are very sorry for our negligence of this part, 7nm is the particle mean size

Section 2.3. How many replicate specimens were prepared and tested?

Response 7: Seven NSCFRC mixtures total 252 specimens and each different concrete 36 specimens were prepared and tested: For each different concrete 27 specimens included 18 cubic specimens and 9 prismatic specimens were heated by box type resistance furnace at 375, 575 and 775°C. According to GB/T50081-2016, 3 cubic specimens for compressive strength test, 3 cubic specimens for splitting tensile strength and 3 prismatic specimens for flexure strength test, the results are averaged.

We have re-written this part in line 110-121

Section 3.1.1, 3.1.2, and 3.1.3: please comment on the relative statistical variability in the data. Based on the results in Table 1, considering the standard deviation, are the results statistically different (for instance, according to a t-test with 95% confidence)? Including a discussion of the statistical variability would significantly bolster the arguments and the conclusions. 

Response 8:  We performed a Hypothesis test on all the data in Table 3 according to one sample t-test with 95% confidence, at the 0.05 level, the population mean is NOT significantly different with the test mean. The experimental results were statistically acceptable: according to Table 3, most of relative standard deviations are below 10%. We added this discuss in line145-148, line 177-181 and line 206-210.

Or we modify Figure 1 to the following form:

Section 3.1.4. The discussion on the porosity derived by SEM imaging of fracture surface should be revised. The authors are drawing too many conclusions based on one image. Yes, the SEM image qualitatively shows more pores, but are the authors certain that what they are interpreting as pores is not a result of the testing? Consider revising this discussion, or provide more definitive analysis of the porosity, such as by MIP, nitrogen adsorption, or image analysis of backscattered electron images of polished sections.

Response 9: According to the Reviewer’s suggestion, we deleted the part about pore analysis in 3.1.4

Section 3.2, 3.3, and 3.4. Same comment as above regarding statistical analysis. By the variability reported in Table 4, 5, and 6, are the results statistically significant? Did nanosilica or fibers have a statistically relevant effect? Consider at least a t-test with 95% confidence. 

Response 10: We also performed a Hypothesis test on all the data in Table 4, 5 and 6 according to one sample t-test with 95% confidence, at the 0.05 level, the population mean is NOT significantly different with the test mean. the standard deviation of the results indicates that the data is more discrete as the temperature increases, but there is no evidence that nanosilica or carbon fiber has a statistically relevant effect.

We added this discussion in line 254-257, line 297-302 and line 339-341.

Equation 2. There is a missing parenthesis in the equation.

Response 11: We corrected this mistake, and added he meaning of the variables.

Section 3.5. "The decomposition of Ca(OH)2 has no critical influence on the reduction of concrete strength." If Ca(OH)2 converts to CaO + H2O, doesn't CaO have a lower unit volume than Ca(OH)2? If so, decomposition of Ca(OH)2 should increase porosity and therefore reduce strength.

Response: According to the Reviewer’s suggestion, we revise this statement as “The decomposition of Ca(OH)2 increase porosity and therefore reduce strength due to CaO have a lower unit volume than Ca(OH)2

-Section 3.5. "C–S–H starts to decompose at around 560°C and it decomposes into b-C2S at around 600–700°C." The authors may wish to check this reference. I've never heard of C-S-H converting to belite. This doesn't make sense to me. 

Response: We deleted “and it decomposes into b-C2S at around 600–700°C”, it’s not necessary

Section 3.5. The authors should also comment on the influence of temperature on the aggregates. If limestone or dolomite is present, then they would undergo decarbonation at higher temperatures. If quartz is present, then it would undergo a phase transition at 573C and result in cracking. Consider reporting on the minerals present in the aggregates and how this is affected by temperature. 

Response 13: According to the Reviewer’s suggestion, we added some discussion as follows:

Line 275-279, “The reason for it may not necessarily be the addition of fibers and NS, but the use of calcareous aggregates, which perform better than siliceous aggregates under high temperature. Quartz content in siliceous aggregates experiments a crystalline transformation from a-quartz to b-quartz at 500-650℃; this transformation entails a volumetric expansion of the aggregates which generates micro-cracks in the concrete matrix and weakens the material.”

Line 382-387, “Furthermore, siliceous aggregates express unfavourable mechanical properties at high temperature compared to the calcareous aggregates, calcareous aggregates decompose at a higher temperature than siliceous aggregates [36]. Quartz content in siliceous aggregates would undergo a phase transition at 573C and result in cracking. This could be used to explain the better performance of the concrete with calcareous aggregates at high temperature in Fig.4b and Fig.5b.”

Section 3.5. "Compared with steel fiber and polypropylene fiber, carbon fiber had clear disadvantages in improving the mechanical properties of concrete after high-temperature." THis sentence is confusing. Did carbon fibers help or not?

-Section 3.5. Discuss what temperatures polypropylene fibers will melt. Are there any temperature-related effects for carbon fibers?

-Conclusions. For Conclusion #5, I am confused with regard to carbon fiber performance at high temperatures. Were carbon fibers useful or not at high temperatures? Didn't melt?

Response 14: Carbon fiber helps the performance of concrete after high temperature. but it is not as good as steel fiber in mechanical properties and not as good as polypropylene fiber in resistance to thermal spalling. We deleted Conclusion #5, and added discussion this as flowing:

Line 393-396 “From the previous section, it can be seen that the NSCFRC has better residual mechanical properties compared to the concrete without fiber and polypropylene fiber reinforced concrete, but lower than steel fiber reinforced concrete. Polypropylene fiber generally has no significant influence on the improvements of residual mechanical properties for concrete after high temperature.”

Line 322-330 “The reason for such improvements could be attributed to the calcareous aggregates and fiber, the testing temperatures are not high enough to allow carbon fiber steel fiber to be melted so that its ductility could effectively contribute to concrete resisting the failure under tension. Experimental results were similar to the prediction in Gao curve for HSC with hybrid fiber below 600℃. The reason for it may NSC and NSCFRC had less internal porosity. However, at 775 ℃, the residual tensile strength of NSCFRC is lower than Gao curve, because carbon fiber has a deterioration of mechanical properties due to oxidation, and it cannot provide sufficient tensile strength above 600℃. Therefore, steel fiber reinforced concrete has the highest residual tensile strength at 800 ℃.”

Line 404-408 “Carbon fiber has a high melting point and a burning point (>1000℃), the testing temperatures are not high enough to allow carbon fiber to be melted so that its ductility could effectively contribute to concrete resisting the failure under tension. However, carbon fiber has a deterioration of mechanical properties due to oxidation above 600℃.”

Special thanks to you for your good comments. 

We tried our best to improve the manuscript and made some changes in the manuscript.  These changes will not influence the content and framework of the paper. And here we did not list the changes but marked in revised paper.

We appreciate for Reviewers’ warm work earnestly, and hope that the correction will meet with approval.

Once again, thank you very much for your comments and suggestions,

Kind regards,

Yours sincerely

Wu Linsong

Reviewer 2 Report

The authors present a comprehensive experimental study about the mechanical properties of NSCRFC mixtures at elevated temperatures. However, the structure of the manuscript is not always clear and sometimes confusing and should therefore by improved. Also, some experimental details are missing.

Abstract:

For the sake of completeness, please provide the whole phrase for "SEM".

Introduction:

Please provide further references for the fact that carbon fiber concrete has been "widely" studied as a smart material, i.e. mention some applications for that. Please explain "finishability".

Please provide an overview about the structure of the paper.

Experimental program:

Line 74: MPa, GPa (instead of Mpa, Gpa) The structure of this section is confusing. Please restructure it. Experimental results of NSCFRC are mentioned before the mixture composition and the testing details are explained. Figure 1 is not clear at all and should be presented at a later stage because the necessary information are provided later. Please provide further details on the used materials (deliverer / manufacturer). Line 104: Why did you heat up the specimens? Which specimens exactly were heated? Line 109: Please provide further details about the SEM investigations (which microscope was used?).

Results and discussion:

Line 120: Why are the compressive strength results shown in Table 3 and Figure 1? Please limit it to one representation. (also for the other mechanical quantities)

Line 202: replace bubbles by pores.

Line 239: Please provide the derivation of this equation and the meaning of the variables. (the same for the subsequent equations)

Line 293: The explanations are too late because the reader does not expect them there.

Author Response

Dear Reviewers:

    Thank you for your letter and for the reviewers’ comments concerning our manuscript entitled “Mechanical Properties of Nano SiO2 and Carbon Fiber Reinforced Concrete after Exposure to High Temperatures” (ID: materials-622485). Those comments are all valuable and very helpful for revising and improving our paper, as well as the important guiding significance to our researches. We have studied comments carefully and have made correction which we hope meet with approval. Revised portion a using the "Track Changes" function in paper. The main corrections in the paper and the responds to the reviewer’s comments are as flowing:

Abstract: For the sake of completeness, please provide the whole phrase for "SEM".

Response 1: Line 12, the “SEM” was corrected as “Scanning Electron Microscopy (SEM)”

Introduction:

Please provide further references for the fact that carbon fiber concrete has been "widely" studied as a smart material, i.e. mention some applications for that. Please explain "finishability".

Response 2: Line 34, Reference: “41.  Ding Y, Han Z, Zhang Y, et al. Concrete with triphasic conductive materials for self-monitoring of cracking development subjected to flexure[J]. Composite Structures, 2016, 138: 184-191. 42.   Ding Y, Huang Y, Zhang Y, et al. Self-monitoring of freeze–thaw damage using triphasic electric conductive concrete[J]. Construction and Building Materials, 2015, 101: 440-446.” was added

And "finishability" refers to the defects of carbon fiber concrete that can be easily repaired after construction

Please provide an overview about the structure of the paper. Experimental program:

Line 74: MPa, GPa (instead of Mpa, Gpa) The structure of this section is confusing. Please restructure it. Experimental results of NSCFRC are mentioned before the mixture composition and the testing details are explained. Figure 1 is not clear at all and should be presented at a later stage because the necessary information are provided later. Please provide further details on the used materials (deliverer / manufacturer). Line 104: Why did you heat up the specimens? Which specimens exactly were heated? Line 109: Please provide further details about the SEM investigations (which microscope was used?).

Response 4: We have re-written this part according to the Reviewer’s suggestion:

In 2.1. Materials and Mixture Proportions, we added the manufacturer of most materials; and deleted “fig.1”, this is our negligence;

In 2.2 Preparation, in order to determine the range of carbon fiber content, we did some preliminary experiments, but this is not necessary, so we deleted line 104-106“Pre-experiments show that when the carbon fiber exceeds 0.35vol %, it will be agglomeration in the concrete and cannot be dispersed evenly”

In 2.3 Testing, We have re-written this part “For each different concrete 36 specimens were prepared and tested: 24 cubic specimens (150*150*150 mm) for the compression test and splitting tension test, and 12 prismatic specimens (100*100*400 mm) were made for the flexure test. The objective of this work is aimed at the evaluation of the mechanical properties of NSCFRC after high temperature exposure. For this purpose the following tests were made under four different conditions (25, 375, 575 and 775°C). For each different concrete 27 specimens included 18 cubic specimens and 9 prismatic specimen were heated by box type resistance furnace at 375, 575 and 775°C. Once the target temperature had been reached, it was maintained for 120 mins in order to obtain a homogeneous thermal state within the specimens. Afterwards, the oven was shut down and the samples were kept inside for natural cooling. All samples were tested after cooling up to ambient temperature.”

Results and discussion:

Line 120: Why are the compressive strength results shown in Table 3 and Figure 1? Please limit it to one representation. (also for the other mechanical quantities)

Response: We have made correction according to the Reviewer’s comments.

Line 202: replace bubbles by pores.

Response: Other reviewers suggested that the evidence of using SEM to analyze the pore structure of concrete was insufficient, so we deleted the part about pore analysis in 3.1.4

Line 239: Please provide the derivation of this equation and the meaning of the variables. (the same for the subsequent equations)

Response: The experimental results were worked to design equations by the means of regression analyses. R2 is coefficient of determination,

In Eqn. (1),where  is the residual compressive strength of concrete after exposure to high temperature T;  is the compressive strength of concrete before exposure to high temperature; T is the elevated temperature experienced by the concrete (unit: ℃)

In Eqn. (2) ,where  is the residual tensile strength of concrete after exposure to high temperature T;  is the tensile strength of concrete before exposure to high temperature;

Line 293: The explanations are too late because the reader does not expect them there.

Response: We added discussions in some sections based on the reviewer's recommendations.

In 3.2, line 273-283, we added discussions “It can be observed that NSCFRC gave better residual strength than the EC-2 and polypropylene fibers reinforced concrete from Aslani et al and Ding et al curve, included PC (0% CFs and 0% NS), The reason for it may not necessarily be the addition of fibers and NS, but the use of calcareous aggregates, which perform better than siliceous aggregates under high temperature. Quartz content in siliceous aggregates experiments a crystalline transformation from a-quartz to b-quartz at 500-650℃; this transformation entails a volumetric expansion of the aggregates which generates micro-cracks in the concrete matrix and weakens the material. Polypropylene fiber generally has no significant influence on the improvements of residual compressive strength for concrete after heating to high temperature [32, 33]. The residual compressive strength reduces when the temperature is above 400 ℃ due to the fiber has been melted up at such high temperature and the pores left are disadvantage for the performance of concrete.”

In 3.3, line 322-330, we added discussions “The reason for such improvements could be attributed to the calcareous aggregates and fiber, the testing temperatures are not high enough to allow carbon fiber steel fiber to be melted so that its ductility could effectively contribute to concrete resisting the failure under tension. Experimental results were similar to the prediction in Gao curve for HSC with hybrid fiber below 600℃. The reason for it may NSC and NSCFRC had less internal porosity. However, at 775 ℃, the residual tensile strength of NSCFRC is lower than Gao curve, because carbon fiber has a deterioration of mechanical properties due to oxidation, and it cannot provide sufficient tensile strength above 600℃. Therefore, steel fiber reinforced concrete has the highest residual tensile strength at 800 ℃.”

Special thanks to you for your good comments. 

We tried our best to improve the manuscript and made some changes in the manuscript.  These changes will not influence the content and framework of the paper. And here we did not list the changes but marked in revised paper.

We appreciate for Reviewers’ warm work earnestly, and hope that the correction will meet with approval.

Once again, thank you very much for your comments and suggestions.

Kind regards,

Yours sincerely

Wu Linsong

Round 2

Reviewer 2 Report

The authors improved the quality of the manuscript and considered the comments of the reviewers carefully. Particularly, the description of the experimental methods is re-structured and improved. However, a few minor corrections are suggested in order to clarify further things. Furthermore, English editing should be performed in order to improve the language style.

Introduction: Please provide a short paragraph where you describe the structure of the paper. It helps the reader to get an overview about the content of the paper.

Please replace "air bubbles" by "pores" within the whole article.

Typo: Line 259: "experiences" instead of "experiments"

Line 315: Paragraph missing

Author Response

Dear Reviewers:

    Thank you for your letter and for the reviewers’ comments concerning our manuscript entitled “Mechanical Properties of Nano SiO2 and Carbon Fiber Reinforced Concrete after Exposure to High Temperatures” (ID: materials-622485). Those comments are all valuable and very helpful for revising and improving our paper, as well as the important guiding significance to our researches. We have studied comments carefully and have made correction which we hope meet with approval. Revised portion a using the "Track Changes" function in paper. The main corrections in the paper and the responds to the reviewer’s comments are as flowing:

Introduction: Please provide a short paragraph where you describe the structure of the paper. It helps the reader to get an overview about the content of the paper.

Response: According to the Reviewer’s suggestion, we added a short paragraph in Line 72-77: “The study is realized in two stages. Firstly, this paper supplies results on the mechanical properties of seven mixtures at room temperature. The effect of carbon fiber content and NS content on the mechanical properties and microstructure of concrete was revealed. Secondly, seven mixtures heated to 375℃, 575℃, 775°C and natural cooling. The evolution of residual mechanical properties with temperature were investigated in order to explain the influence of carbon fiber and NS on the mechanical properties of concrete after high temperature.”

Please replace "air bubbles" by "pores" within the whole article.

Typo: Line 259: "experiences" instead of "experiments"

Line 315: Paragraph missing

Response: We are very sorry for our negligence and have made correction according to the Reviewer’s comments

Special thanks to you for your good comments. 

We tried our best to improve the manuscript and made some changes in the manuscript.  These changes will not influence the content and framework of the paper. And here we did not list the changes but marked in revised paper.

We appreciate for Reviewers’ warm work earnestly, and hope that the correction will meet with approval.

Once again, thank you very much for your comments and suggestions.

Kind regards,

Yours sincerely

Wu Linsong